Apparent source levels and active communication space of whistles of free-ranging Indo-Pacific humpback dolphins (Sousa chinensis) in the Pearl River Estuary and Beibu Gulf, China

Wang Zhi-Tao 1 2 3 9
W.L. Au Whitlow 3
Rendell Luke 4
Wang Ke-Xiong 1 wangk@ihb.ac.cn
Wu Hai-Ping 5
Wu Yu-Ping 6
Liu Jian-Chang 7
Duan Guo-Qin 8
Cao Han-Jiang 8
Wang Ding 1 wangd@ihb.ac.cn
1 The Key Laboratory of Aquatic Biodiversity and Conservation of the Chinese Academy of Sciences, Institute of Hydrobiology of the Chinese Academy of Sciences , Wuhan, Hubei , China
2 University of Chinese Academy of Sciences , Beijing , China
3 Marine Mammal Research Program, Hawaii Institute of Marine Biology, University of Hawaii , Hawaii, HI , United States of America
4 Sea Mammal Research Unit, School of Biology, University of St. Andrews , Fife , United Kingdom
5 School of Marine Sciences, Qinzhou University , Guangxi , China
6 School of Marine Sciences, Sun Yat-Sen University , Guangzhou , China
7 Transport Planning and Research Institute, Ministry of Transport , Guangzhou , China
8 Hongkong-Zhuhai-Macao Bridge Authority , Guangzhou , China
9 Current affiliation: Division of Marine Science and Conservation, Nicholas School of the Environment, Duke University , Beaufort, NC , United States of America
Pawlik Joseph
Electronic publication date: 2016 Feb 15
Publication date: 2016
Volume: 4
Electronic Location ID: e1695
Received 2015 Nov 25; Accepted 2016 Jan 26
Copyright: ©2016 Wang et al.
Copyright year: 2016
Copyright holder: Wang et al.
License: This is an open access article distributed under the terms of the Creative Commons Attribution License, which permits unrestricted use, distribution, reproduction and adaptation in any medium and for any purpose provided that it is properly attributed. For attribution, the original author(s), title, publication source (PeerJ) and either DOI or URL of the article must be cited.
License URL: https://creativecommons.org/licenses/by/4.0/

Keywords: Active communication space, Pearl River Estuary, Sound propagation model, Whistles, Indo-Pacific Humpback dolphins, Hydrophone arrays, Beibu Gulf, Apparent source level, Sousa chinensis

Funding: National Natural Science Foundation (NNSF) of China 31070347 Ministry of Science and Technology of China 2011BAG07B05-3 Knowledge Innovation Program of the Chinese Academy of Sciences KSCX2-EW-Z-4 Special Fund for Agro-scientific Research in the Public Interest of the Ministry of Agriculture of China 201203086 State Oceanic Administration of China 201105011-3 NNSF of China 31170501 China Scholarship Council (2014)3026 Grants for this study was provided by the National Natural Science Foundation (NNSF) of China (Grant No.31070347), the Ministry of Science and Technology of China (Grant No. 2011BAG07B05-3), the Knowledge Innovation Program of the Chinese Academy of Sciences (Grant No. KSCX2-EW-Z-4) and the Special Fund for Agro-scientific Research in the Public Interest of the Ministry of Agriculture of China (Grant No. 201203086) to DW, the State Oceanic Administration of China (Grant No. 201105011-3) and NNSF of China (Grant No. 31170501) to KXW and the China Scholarship Council (Grant No. (2014)3026) to ZTW. The funders had no role in study design, data collection and analysis, decision to publish, or preparation of the manuscript.

==============================
Background. Knowledge of species-specific vocalization characteristics and their associated active communication space, the effective range over which a communication signal can be detected by a conspecific, is critical for understanding the impacts of underwater acoustic pollution, as well as other threats.

Methods. We used a two-dimensional cross-shaped hydrophone array system to record the whistles of free-ranging Indo-Pacific humpback dolphins (Sousa chinensis) in shallow-water environments of the Pearl River Estuary (PRE) and Beibu Gulf (BG), China. Using hyperbolic position fixing, which exploits time differences of arrival of a signal between pairs of hydrophone receivers, we obtained source location estimates for whistles with good signal-to-noise ratio (SNR ≥10 dB) and not polluted by other sounds and back-calculated their apparent source levels (ASL). Combining with the masking levels (including simultaneous noise levels, masking tonal threshold, and the Sousa auditory threshold) and the custom made site-specific sound propagation models, we further estimated their active communication space (ACS).

Results. Humpback dolphins produced whistles with average root-mean-square ASL of 138.5 ± 6.8 (mean ± standard deviation) and 137.2 ± 7.0 dB re 1 µPa in PRE (N = 33) and BG (N = 209), respectively. We found statistically significant differences in ASLs among different whistle contour types. The mean and maximum ACS of whistles were estimated to be 14.7 ± 2.6 (median ± quartile deviation) and 17.1± 3.5 m in PRE, and 34.2 ± 9.5 and 43.5 ± 12.2 m in BG. Using just the auditory threshold as the masking level produced the mean and maximum ACSat of 24.3 ± 4.8 and 35.7 ± 4.6 m for PRE, and 60.7 ± 18.1 and 74.3 ± 25.3 m for BG. The small ACSs were due to the high ambient noise level. Significant differences in ACSs were also observed among different whistle contour types.

Discussion. Besides shedding some light for evaluating appropriate noise exposure levels and information for the regulation of underwater acoustic pollution, these baseline data can also be used for aiding the passive acoustic monitoring of dolphin populations, defining the boundaries of separate groups in a more biologically meaningful way during field surveys, and guiding the appropriate approach distance for local dolphin-watching boats and research boat during focal group following.

Introduction

Human activities have profoundly changed the world’s aquatic environment. The International Union for the Conservation of Nature (IUCN) suggests that nearly half of the extant marine mammal species are threatened by two or more human impacts, and that a quarter of marine mammals have been classified as threatened with extinction (Davidson et al., 2012). The Indo-Pacific humpback dolphins (Sousa chinensis, locally called the Chinese white dolphin) is widely distributed throughout shallow, coastal waters from eastern India in the west to the Southern China Sea in the east and throughout Southeast Asia (Jefferson & Rosenbaum, 2014; Reeves et al., 2008). However, marine mammal species occurring in coastal areas are most susceptible to risk, and the coastal distribution of the humpback dolphins make it highly vulnerable to the impact of human activity (Davidson et al., 2012). Its conservation status was categorized as Near Threatened by the IUCN Red List of Threatened Species (Reeves et al., 2008) and as a Grade One National Key Protected Animal in China. Five resident populations of Indo-Pacific humpback dolphins have been identified in Chinese coastal waters: the Pearl River Estuary (PRE) (Chen et al., 2010), Leizhou Bay (Xu et al., 2015) of Guangdong, the Beibu Gulf (BG) of Guangxi (Chen et al., 2009; Pan et al., 2006), Xiamen harbor of Fujian (Chen et al., 2009), and the West coast of Taiwan (Wang et al., 2012).

The PRE region (Fig. 1) is among the most economically developed regions in China (Yeung & Shen, 2008) and also home the world’s largest known population of humpback dolphins (Chen et al., 2010; Preen, 2004), with the population size estimated to be over 2,500 (CVs: 19–89%) (Chen et al., 2010). The BG region (Fig. 1) is, in comparison, relatively undeveloped, with a smaller human population, and the humpback dolphin population there was estimated to be 251 (95% CI [136–794]) (Chen et al., 2009; Pan et al., 2006). The concern about the effects of anthropogenic noise on aquatic life is growing world widely (Popper & Hawkins, 2012), and economic growth in China has been accelerating human damage to coastal ecosystems (He et al., 2014). The recent construction of the Hongkong-Zhuhai-Macao bridge (Wang et al., 2014b), the Zhuhai wind-farm project in Pearl River Estuary, and the flourishing year round dolphin-watching industry in Beibu Gulf (Wang et al., 2013) all have potentially adverse effects on aquatic life. Pile-driving is likely to cause acoustic disturbance (Wang et al., 2014b), and the intense dolphin-watching industry make the dolphin susceptible to close approaches by high-speed dolphin-watching vessels. High-speed vessels can seriously affect the dolphins’ natural behavior (Ng & Leung, 2003), introduce masking noise (Sims, Hung & Wursig, 2012a), and cause injury or even death (Jefferson, 2000) to resident cetaceans. Hence, concerns regarding the conservation of these Chinese white dolphin populations are increasing.

Figure 1 Map of the study area.

Acoustic recordings of underwater sounds produced by humpback dolphins were made in Pearl River Estuary and Beibu Gulf. Dashed line area shows the sound recording region.

Marine mammals, especially cetaceans, have evolved sophisticated sound production and reception mechanisms to aid in meeting their requirements for a series of vital processes, including communication, navigation, and foraging (Au, 1993; Au & Hastings, 2008; Surlykke et al., 2014). Dolphins use frequency modulated narrowband sounds, also called whistles, for communication with conspecifics (Janik, 2000b; Janik & Slater, 1998). Both whistle source level (SL), defined as the amplitude at 1 m from the animal on the acoustic axis (Janik, 2000a) and its associated active communication space, the effective range over which a communication signal can be detected by a conspecific (Marten & Marler, 1977; Tervo et al., 2012) are fundamental parameters in animal communication systems. The source level is important because it can provide information on the biological ambient noise caused by conspecifics to which an animal is exposed (Janik, 2000a), which can shed some light on evaluating the appropriate exposure level of dolphins to anthropogenic noise. Knowledge of the statistical distribution of whistle source levels can help in planning passive acoustic monitoring studies of habitat use, as well as abundance estimates (Frankel et al., 2014; Širović, Hildebrand & Wiggins, 2007). However, the distance commonly used to identify dolphins as members of a group was either the ‘10-m chain rule’ (any individuals considered part of the same group were within 10 m of at least one other member of the group, regardless of behavior) (Acevedo-Gutiérrez, 2002; Acevedo-Gutiérrez & Stienessen, 2004; Connor, Smolker & Bejder, 2006; Quick & Janik, 2008; Quick & Janik, 2012; Smolker et al., 1992) or a radius of 100 m (a collection of individuals within which no dolphins were separated by greater than 100 m) (Barco et al., 1999; Lewis, Wartzok & Heithaus, 2011), which may not be biologically meaningful. In conjunction with passive acoustic localization, many recorded whistles from a dolphin focal group (defined by 10-m chain rule) were confirmed to be produced by non-focal groups nearby, rather than the defined focal group (Quick & Janik, 2008). Also, the estimated whistle active space in previous studies of odontocetes were mismatched with, and always greater than, the separation distances commonly used to define the boundary of separate groups (Janik, 2000a; Miller, 2006; Quintana-Rizzo, Mann & Wells, 2006). Additionally, with the increasing threaten of the acoustic masking in marine ecosystems by anthropogenic noise (Clark et al., 2009), the active communication space can help to define the boundary of separate dolphin groups in a more biologically meaningful way.

Humpback dolphin can emit pulsed sound with a peak frequency of 114 ± 12 kHz and an apparent source level of 199 ± 3 dB re 1 µPa @ 1 m (peak-to-peak) (Freitas et al., 2015). Also, they can produce whistles with fundamental frequencies averaged 6.4 kHz, and minimum and maximum fundamental frequencies averaged 5.1 kHz and 7.7 kHz, respectively (Wang et al., 2013). Although S. chinensis is a common species in many waters, information about its vocal behavior remains sparse (Hoffman et al., 2015; Kimura et al., 2014; Li et al., 2012; Wang et al., 2013; Wang et al., 2015). The regulation of underwater acoustic pollution is currently constrained by sparse data, especially the scarcity of quantitative data on animal vocalization characteristics and effects of anthropogenic noise on the biological functions, such as acoustically mediated social interactions (NRC, 2005). In order to avoid or to mitigate the possible detrimental impact and to better protect these Sousa populations, basic acoustic information is needed.

While the apparent source level of whistles, defined as the back-calculated sound pressure level at 1 m distance from the sound source at an unknown angle from the acoustic axis (Jensen et al., 2009b), and its active communication space were estimated in many cetaceans, such as in bottlenose dolphin (Tursiops truncatus) (Jensen et al., 2012) and in white-beaked dolphins (Lagenorhynchus albirostris) (Rasmussen et al., 2006), relevant information is barely known in humpback dolphin. In this study, by using passive acoustic localization, the apparent source level of whistles produced by free-ranging S. chinensis in Pearl River Estuary and Beibu Gulf were measured. The active communication space of whistles were further estimated by integrating whistle source parameters, real-time measurements of environmental background noise spectrum levels and by modeling of the sound propagation loss for the habitat in question with animal physiological hearing capabilities and critical ratios.

Methods

Data collection

Acoustic recordings were made during June–July, 2014, in PRE (22°06′–22°11′S; 113°40′–113°45′E) and August 2014 in BG (21°30′–21°37′S; 108°40′–108°58′E), China (Fig. 1). Surveys were conducted from a 7.5 m recreational powerboat with a 140 hp outboard engine in PRE or a 6.8 m dolphin-watching vessel powered by 40 hp outboard engine in BG under Beaufort sea states ≤3 (on a scale of 12) with a randomly selected route rather than structured transects.

When a group of dolphins was sighted and the majority of whose members were engaged in slow or moderate movements (resting, milling, socializing or feeding) (Hawkins & Gartside, 2009), the vessel moved position to the side of the dolphin group. Groups were defined by the ‘10 m chain rule’ (Quick & Janik, 2012). If the dolphin group was traveling fast (Hawkins & Gartside, 2009), the boat would move swiftly ahead of their moving direction to await them passing by. During sound recording, the vessel’s engine was turned off. For each animal group, the GPS time, location (latitude and longitude), dolphin species, and behavior (traveling, socializing, milling, resting, and feeding) (Hawkins & Gartside, 2009) were recorded. The water depth and water quality, including temperature, salinity, and pH, were measured with a Horiba Multi-parameter Water Quality Monitoring System (model W-22XD; Horiba, Ltd., Kyoto, Japan) for sound propagation modeling. Recording was stopped when none of the dolphins of a group were within 50 m to the hydrophone arrays.

The two-dimensional cross-shaped array consisted of five Reson piezoelectric hydrophones, one in the middle and four on each end of the arms (model TC-4013, frequency range 1 Hz–170 kHz, sensitivity: −211 dB ± 3 dB re 1 V/µPa; Reson Inc., Slangerup, Denmark) (Fig. 2). Each hydrophone was equipped with a 1 MHz bandwidth Reson EC6081 voltage pre-amplifier with a band-pass filter (model VP2000, pass-band 0.1 to either 100 kHz or 250 kHz depending on sampling rate). The EC6081 employ the first order filters (one pole), which was a filter slope of 6 dB/octave in frequency. The hydrophones were connected via a 16-channel synchronized analogue-to-digital (A/D) converter to a laptop computer running LabVIEW 2011 SP1 software (National Instruments (NI), Austin, TX, USA). The A/D converter consisted of four high-speed, 16 bit resolution, data acquisition (DAQ) modules (NI 9223), incorporated in a compact DAQ four-slot USB chassis (NI cDAQ-9174). Each NI 9223 was a four-channel simultaneous A/D converter with a sample rate up to 1 MHz for each channel. Both VP2000 amplifier and NI cDAQ-9174 were powered by external battery packs.

Figure 2 Schematic of experimental apparatus and the array design.

Acoustic signals was picked by the hydrophones and conditioned by the amplifier and filtered before storage into the PC via the DAQ systems. Distance between H1, H2, H3, H4, and H5 was 1.47 m and 1.54 m for Pearl River Estuary and Beibu Gulf, respectively. Distance between H1, H2, H3, and H4 was 2.08 m and 2.18 m for Pearl River Estuary and Beibu Gulf, respectively. The inset shows a detailed view of the hydrophone array.

A steel bracket was used to fix the distance between hydrophones. The bracket was made from a stainless cylinder-shaped bar with a cross structure as its backbone (bar diameter: 2.5 cm) and a reinforced stainless bar (bar diameter: 2 cm) at each quadrant (Fig. 2). A 5 cm extending bar (bar diameter: 0.3 cm) was affixed perpendicularly to the bracket plane at the center and end of each arms to mount the hydrophones, to minimize the interference of the bar to the sound (including reflection and/or shadowing) (Fig. 2). Inter-hydrophone distance along the backbone structure of the bracket was 1.47 m in PRE and 1.54 m in BG.

During the sound recording, the hydrophone arrays were deployed from the side of the boat so that the plate was in the horizontal plane at a depth of 1 m. Floats and attached weights limited array movement to reduce noise due to water flow (Fig. 2). The acoustic data were stored directly on the hard drive of a computer in technical data management streaming (TDMS) format and sampled at a rate of either 200 kHz or 512.828 kHz, giving a Nyquist frequency of 100 kHz and 256.414 kHz, respectively. The presence of signals was monitored in real-time by using both the PC screen for waveforms monitoring and a headphone connected to the center hydrophone. To minimize the chance of missing good signal, a three second pre-recording buffer was employed. Upon detecting a signal, a manual trigger was used to initiated a recording with the buffer included.

The Reson hydrophones were calibrated prior to shipment from the factory (Fig. S1). The remaining components of the recording system, including the amplifier, filter, A/D converter and laptops, were calibrated in the lab prior to the field survey by inputting a calibration signal generated by an OKI underwater sound level meter (model SW1020; OKI Electric Industry Co., LTD., Tokyo, Japan). Signal flow was also simultaneously monitored with an oscilloscope (model TDS1002C; Tektronix Inc., Beaverton, OR, USA). The noise floor of the recording system was about 65 and 55 dB re 1 µPa2/Hz at 100 Hz and 1 kHz, respectively, and flat at about 50 dB re 1 µPa2/Hz between 10 kHz and 100 kHz, which were lower than the ambient noise level at sea state 0 in our study (Fig. S1), and suitable for noise monitoring.

Sound propagation modeling

Multi-path propagation is inevitable in shallow waters, as bottom and surface reflections interfere with the signal propagation in a direct path. Following standard sound propagation theory (Au & Hastings, 2008; Aubauer, Lammers & Au, 2000; Urick, 1983), a custom-compiled sound propagation model (File S1) targeted on the impact of multi-path propagation on the original signals and took into account the hydrophone-animal geometry (such as animal depth, hydrophone depth, distance between hydrophone and animal) and site specific environment and bathymetry characteristics (such as water depth and bottom sediment contents) was adopted for this study (Fig. 3).

Figure 3 Schematic of multipath propagation.

The dw, da and dh were the depth of the water, the animal, and the receiving hydrophone, respectively. “A” denotes the animal location, and “H” denotes the hydrophone, Aa was the horizontal separation distance between the animal and the hydrophone, r0 was the direct signal propagation path, rs(m) and rb(m) were the signal propagation lengths for multipath propagation signal with a total number of m reflection points and the initial reflection point at the air–water and water–bottom interface, respectively, θs(m) and Φs(m) were the incident (same as reflected) and transmitted angle, respectively, for multipath propagation signal with a total number of m reflection points and the initial reflection point at the air–water interface, θb(m) and Φb(m) were the incident (same as reflected) and transmitted angle, respectively, for multipath propagation signal with a total number of m reflection points and the initial reflection point at the water–bottom interface, hs(m) and hb(m) were the vertical propagation length of the multipath propagation signal with a total number of m reflection points and the initial reflection point at the air–water and water–bottom interface, respectively, by referencing the animal location. The insets show the sound transmission at the air–water interface and at the water–bottom interface, respectively.

Since the energy flux density (EFD, dB re 1 µPa2 s) is more meaningful in situations where considerable signal distortion occurs during propagation (Urick, 1983), the estimated transmission loss (TL) for each location was subsequently derived from the difference from the energy flux density of the received signal (EFDr) and the energy flux density at the signal source (EFDs) by the equation: (1) TL=EFDr−EFDs

(2) EFD=10×log10∫0Tp2t∕p ref12dt

where p(t) is the sound pressure in µPa, then pref1 was 1 µPa2 s. For each hydrophone and animal depth combination at a given water depth, the above obtained transmission losses, at varied separation distances between the hydrophone and animal were fitted to a geometric spreading loss model to estimate the environment–dependent transmission loss coefficient by the equation: (3) TL=k×log10r∕r0

where k was the transmission loss coefficient, r was the distance between the animal and hydrophone, ro was the reference range set as 1 m (Fig. 4). Frequency-dependent absorption was ignored here, since the sound absorption losses for a standard Sousa whistle, with a mean fundamental frequency of 6.35 kHz (Wang et al., 2013) as a function of site specific temperature and pressure at the PRE and BG were 0.31 and 0.30 dB/km, respectively, according to the Fisher and Simmons equation (Fisher & Simmons, 1977), and would be negligible over the ranges at which we actually recorded signals.

Figure 4 Sound transmission loss coefficient as a function of animal depth and distance between hydrophone and animals at given hydrophone and water depth.

The blue curve was the modeled transmission loss of the whistle with a peak frequency of 6.6 kHz (see spectrogram in Fig. 5) at water depth of 4.5 m with hydrophone at 1 m depth and animal located at (A) surface, (B) middle section and (C) bottom of the water in Beibu Gulf. The red curve in each graph represents logarithmic curve fit of the blue curve.

Figure 5 Schematic of acoustic localization of humpback dolphins whistle.

(A) oscillograms of same signal received at four different hydrophones (H1, H2, H3, and H4). Cross-correlation was shown in (B), and legends on the top left corner of each panel indicate which two hydrophones have been cross-correlated. The peak of each correlation function corresponds to time differences in time of arrival of whistles in the front hydrophone minus that of the later one for the compared hydrophones. Hyperbola fixing (in C) and legends next to each hyperbola indicate which hydrophone pair it corresponds to. Points of intersection of hyperbolae indicate position of sound source. Closed blue circle (in C) indicates position of hydrophone arrays. Point (0, 0) was located at the center of the acoustic array. The slide on the top right corner of (C) optimize estimated depth of the animal.

Acoustic data analysis

The peripheral four hydrophone channels were used for the acoustic localization of phonation animals, and the center hydrophone channel was used for detailed whistle characteristic measurement. Raven Pro Bioacoustics Software (version 1.4; Cornell Laboratory of Ornithology, NY, USA) was used to analyze the acoustic data in spectrogram (window type: Hann windows; FFT size: 8,192 and 16,384 samples for sampling frequencies of 200 and 512 kHz, respectively; frame overlapping: 80%). Only whistles with good signal-to-noise ratios (SNR ≥ 10 dB) on all five hydrophones and satisfying the criteria of no overlapping echolocation signal or whistles from different individuals were analyzed. In order to make the data more independent and to reduce the possibility of using multiple whistles from the same animal, for each dolphin encounter, we extracted only one signal for each whistle tonal type (Wang et al., 2013) for further analyzing.

Acoustic localization

Passive acoustic localization of vocalizing animals based on differences in the time of arrival of the same sound between all pairs of hydrophone receivers is a well-established technique (Au & Benoit-Bird, 2003; Janik, 2000b; Jensen et al., 2009a; Spiesberger, 1997; Spiesberger & Fristrup, 1990; Wahlberg, Møhl & Madsen, 2001; Watkins & Schevill , 1972). In this study, a custom-written package based on Matlab software (version R2010b; The Mathworks, Inc., Natick, MA, USA), named TOADY (King, Harley & Janik, 2014; Quick, Rendell & Janik, 2008; Quick & Janik, 2012; Quick & Janik, 2008; Schulz, Whitehead & Rendell, 2006; Schulz et al., 2008) was adopted for localizing phonating animals. The time delays were preserved on the simultaneous multi-track recording of signal input from all hydrophones. Signal waveforms from the different recording channels were cross-correlated to determine the difference in arrival time of a sound at each hydrophone pair. Before cross-correlation processing, a digital high-pass filter set to start rolling off just below the minimum frequency of the fundamental frequency contour of each whistle was used to eliminate any low-frequency background noise interference. The position of the largest peak in the resulting cross-correlation vector represents the amount by which the two signals are offset in time (Hayes et al., 2000). Signals with more than one equivalent peak and/or low cross-correlation maxima were discarded (Lammers & Au, 2003). The time delays were used to generate hyperboloid surfaces of possible source locations.

The standard hyperboloid can be estimated by rotating a standard hyperbola along its transverse axis. In detail, the standard hyperbola can be constructed by equations: (4) x2∕aij2−y2∕bij2=1

(5) aij=c×ti−j∕2

(6) dij=xi−xj2+yi−yj2+zi−zj2

(7) bij=dij∕22−aij2

where (x, y) represent the locus coordinates in two dimensions at the hyperbola located alone the hydrophone array plane, aij and bij represent the distance from the center to either vertex and the length of a segment perpendicular to the transverse axis drawn from each vertex to the asymptotes of the hyperbola between the hydrophone i and j, respectively. The symbols (xi, yi, zi) and (xj, yj, zj) represent the three-dimension coordinates of the hydrophone i and j, respectively, c represents the speed of sound in water in m/s, and ti−j represents the time delay between the hydrophones i and j in seconds. The maximum allowable time delay between a pair of hydrophones in the array is limited to the direct-path propagation time between them (Helble et al., 2015) as: (8) maxti−j=dij∕c

where dij represent the separation of the hydrophone i and j in m. The standard hyperboloid was then rotated and further recast to the center of the spatial geometry of the corresponding array-pair positions.

Once all the hyperboloids were established, contours of the hyperboloids (hyperbolae) at varied assumed animal depths, ranging from the water surface to the bottom set at 0.5 m increments, were displayed in the graphical interface of the TOADY software for visual inspection the hyperbolic fixing (Fig. 5C). Four hydrophones resulted in six hyperbolae and yield four points of intersection (for each independent combination of a hydrophone triad, only two of the three time differences were linearly independent, and all three hyperbolae intersected at a single point) (Laurinolli et al., 2003). The localization accuracy was increased by inclusion of the depth function (Quick, Rendell & Janik, 2008), and animal depth was estimated as that where the surface area of the polygon formed by the hyperbola intersections was minimum (Quick, Rendell & Janik, 2008). The average of the hyperboloid intersections was taken as the best estimate of the sound source’s location (Clark & Ellison, 2000; Laurinolli et al., 2003; Schulz, Whitehead & Rendell, 2006; Schulz et al., 2008).

Ideally, all the four intersections occurred at one point (Fig. 5C). The location error was assessed by a linear error propagation model (Taylor, 1997), and the root-mean-square (rms) location error was estimated using the equation: (9) εrms=εx2+ εy2+ εz2

where εx, εy, and εz are the standard deviation (SD) of the hyperbolae intersections in the zonal, meridional, and vertical directions, respectively (Laurinolli et al., 2003; Schulz et al., 2008; Wahlberg, Møhl & Madsen, 2001).

Signal extraction

Whistles with successful source location estimates were extracted for sound parameter analysis using the center hydrophone channel. The extracted whistle was assigned to one of the following six tonal types based on its fundamental time-frequency contour as: flat, down, rise, U-shape, concave and sine. All tonal types were mutually exclusive (Fig. 6, for detailed definition, see Wang et al., 2013). A three-step procedure was applied to extract the candidate whistles (Fig. 7). A 2-s signal was extracted for each candidate whistle (the whole signal in Figs. 7A and 7B). The actual whistle was subsequently measured from the start and end points of the fundamental contour (Fig. 7C) and further extracted it as the portion containing 98% of the total cumulative energy, which started at the time when 1% of the cumulative signal energy was reached (t1%ce, in Fig. 7E) and ended when 99% of the cumulative signal energy was reached (t99%ce, in Fig. 7E). Whistle duration was derived from the time difference between the 1st and 99th cumulative energy percentiles (in Fig. 7E). A 500 ms ambient noise selection was extracted either before of after (in Fig. 7B) each whistle from the 2-s signal, with a gap of over 0.2 ms from either sides of the whistle fundamental contour (in Fig. 7B). All spectrograms were computed with 25 ms Hann windows (5,000 and 12,820 samples, zero-padded to 8,192 and 16,384 samples for sampling frequencies of 200 and 512 kHz, respectively) for FFT computation with 80% overlap for a temporal resolution of 5 ms and an interpolated spectral frequency resolution of 24.4 and 31.1 Hz, respectively.

Figure 6 Spectrogram of the six whistle tonal types.

Spectrogram configuration (window type: Hanning; temporal grid resolution 5 ms; overlap samples per frame 80%; frequency grid spacing 24.4 Hz; window size 5,000; FFT size 8,192; Nyquist frequency 100 kHz). Note that spectrogram maximum frequency was scaled to 25 kHz for a detailed view of the whistle fundamental frequency.

Figure 7 Three-step whistle extraction.

(A) waveform and (B) spectrogram of the 2 s signal extracted for each whistle. Candidate whistle was extracted from the starting and ending point of the trace of the whistle fundamental frequency contour (in C) and further extracted as the portion containing 98% of the total cumulative energy (between ce1% and ce99% in E), whistle duration was defined as the time between the 1st and 99th cumulative energy percentiles (between t99%ce and t1%ce in E). A 500 ms ambient noise selection was extracted ahead of or following (in A and B) each whistle as the matched noise. Spectrogram configuration (window type: Hanning; temporal grid resolution 5 ms; overlap samples per frame 80%; frequency grid spacing 31.3 Hz; window size 12,821; FFT size 16,384; Nyquist frequency 256.414 kHz). Note that spectrogram maximum frequency was scaled to 20 kHz for a detailed view of the fundamental frequency.

Apparent source levels and source energy flux density

For each whistle, the root-mean-square sound pressure levels (SPLrms, dB re 1 µPa) and energy flux density (EFD) were calculated using the following equations (Au & Hastings, 2008): (10) SPLrms=10×log101∕T×∫0Tp2t∕p ref22dt

where p(t) was the sound pressure in µPa, and pref2 was 1 µPa. SPLrms critically relies upon the signal window size (T) in Eq. (10) (Madsen, 2005). Bottlenose dolphins integrate pure-tone acoustic energy in the same way as humans (Johnson, 1968b), with the integrating time constant for the pure-tone range from 1 kHz to 8 kHz approximately 200 ms (Johnson, 1968b; Plomp & Bouman, 1959). The representative range of the fundamental frequencies of the Sousa whistle averaged at 6.4 kHz with the minimum and maximum fundamental frequency average at 5.1 kHz and 7.7 kHz, respectively (Wang et al., 2013). Here, we assumed that the integration time constant from bottlenose dolphins also applied to Sousa. Both whistles and matched noise samples were consecutively cut into segments of 200 ms with two adjacent slices overlapping by 95%. A measure termed SPLrms200 was taken as maximum SPLrms value from the 200 ms slices of each whistle, and SPLnoi was derived from the average SPLrms value of the 200 ms, slices of each matched noise sample. Absolute pressure levels were derived by incorporating the sensitivity of the hydrophone and the amplifier gain (Au & Hastings, 2008). Apparent source levels (ASLs) and source energy flux density (SEFD) were estimated from the received apparent sound pressure levels and energy flux density by compensating for the transmission loss using the site-specific transmission loss model.

Power spectral density and one-third octave band levels

Power spectral density (dB re 1 µPa2Hz−1), the averaged sound power in each 1 Hz band (Sims et al., 2012b) were calculated using Welch approach for each whistle over its 98% energy windows and their corresponding noise to assess the detailed acoustic energy distribution. Their one-third octave band levels (dB re 1 µPa2) were further calculated to assess how cetaceans auditory systems perceive sound and were impacted by ambient noise (Madsen et al., 2006). All power spectral density and one-third octave band levels were computed with 0.2 s slice window, with 95% overlap between two slices for FFT computation, resulting in an interpolated spectral frequency resolution of 3.05 and 3.91 Hz for sampling frequencies of 200 and 512 kHz, respectively.

Active communication space

Detection of a tonal signal against a continuous broad-band noise background will be effectively masked by only a relatively narrow band of frequencies centered on the tonal stimulus, namely the critical bandwidth (Fletcher, 1940). The critical ratio is another measure of auditory filter width and an indirect method for estimating critical bandwidth (Au & Moore, 1990). At the detection threshold, the signal power equals the noise power, so that the auditory filter width is the ratio of the threshold intensity of a tone over the ambient noise power spectral density at the frequency in question (Fletcher, 1940).

The active communication space is a combined function of the signal source level, the dolphin auditory threshold, the habitat-specific transmission loss, and the masking level at the one third octave band center frequency in question (Janik, 2000a; Jensen et al., 2012; Quintana-Rizzo, Mann & Wells, 2006). The masking level is determined by the noise one-third octave band level or the masked tone threshold, whichever dominated. The masked tone threshold is the sum of the noise power spectral density and the critical ratio at the frequency in question (Janik, 2000a; Jensen et al., 2012; Quintana-Rizzo, Mann & Wells, 2006). The active communication space of each whistle is estimated as the maximum range at which the signal can still be detected in at least one of the one-third octave bands analyzed after accounting for the transmission loss (Miller, 2006). For whistle signals, the one-third octave band that determines the maximum range is always at the peak frequency of the signal one-third octave band levels (Fig. 8).

Figure 8 Schematic for active communication space calculation.

The mean(Sig_TOBL) and mean(Noi_TOBL), surrounded by gray shading of a 95% CI were calculated from a running average of the one-third octave band levels for each whistle and the matched noise, respectively, with step window size of 200 ms and 95% steps overlap, fp was the peak frequency determined by the mean(Sig_TOBL), the max(Sig_TOBL) and mean(Noi_PSD) were calculated from a running maximum one-third octave band levels of whistle and a running average power spectral density of the matched noise, respectively, both with step window size of 200 ms and 95% steps overlap. Sousa audiogram with a frequency span of 500 Hz–38 kHz was obtained by fitting a third-order polynomial curve to the Sousa auditory thresholds between 5 kHz and 38 kHz. Dolphin critical ratio was adopted from Johnson, McManus & Skaar (1989). The inset shows a detailed portion of the max(Sig_TOBL) and mean(Sig_TOBL) at the peak frequency determined by the averaged one-third octave band levels for all the 200 ms slices for each whistle.

The active communication space for each whistle can be modeled by the equations: (11) k×log10meanACS=TL=meanSig_TOBLfp− maxML(fp),AT(fp)

(12) k×log10maxACS=TL=maxSig_TOBLfp− maxMLfp,ATfp

(13) MLfp= maxmeanNoi_TOBLfp,MTTfp

(14) MTT(fp)=meanNoi_PSDfp+CR(fp)

where ACS was the active communication space of the whistle in the near simultaneous ambient noise conditions obtained from the matched noise sample, mean(Sig_TOBL) and max (Sig_TOBL) were the averaged and maximum one-third octave band level for all the 200 ms slices for each whistle, fp was determined by the peak frequency of the averaged one-third octave band levels for all the 200 ms slices for each whistle, mean(Noi_PSD) and mean(Noi_TOBL) were the averaged power spectral density and one-third octave band levels of all the 200 ms slices from the matched noise sample for each whistle, ML was the masking level, MTT was the masked tone threshold, and AT was Sousa auditory threshold. The Sousa audiogram with a frequency span of 500 Hz–38 kHz (which cover the fundamental contour range of Sousa whistles of 520 Hz–33 kHz (Wang et al., 2013)) was estimated by fitting a third-order polynomial curve to the auditory thresholds between 5 kHz and 38 kHz (Li et al., 2012). CR was the dolphin critical ratio (Johnson, McManus & Skaar, 1989), and was obtained by following the equation: (15) CR=19.8+0.075×f1∕2

where CR was in dB and f was the frequency in Hz. The equation was obtained by applying a least-square fit to the bottlenose dolphin critical ratio data (Johnson, 1968a; Moore & Au, 1982).

In cases where the masking level was always higher than the relevant Sousa auditory threshold, i.e., the active communication space was noise-limited, the theoretical active communication space determined by the Sousa auditory threshold alone was also calculated. The active communication space determined by auditory threshold alone (ACSat) can be modeled as: (16) k×log10meanACSat=TL=meanSig_TOBLfp−ATfp

(17) k×log10maxACSat=TL=maxSig_TOBLfp−ATfp.

Statistical analysis

Descriptive statistics of all measured acoustic parameters were obtained and presented in the form of mean, SD, and ranges if they were normal distributed. For those parameters with a grossly skewed distribution, descriptive parameters of median, quartile deviation (QD), 5 percentile (P5), and 95 percentile (P95) were adopted. The Levene’s test for equality of error variances and Kolmogorov–Smirnov goodness-of-fit test were used to analyze homogeneity of variance and the distributions of the data, respectively. Nonparametric statistical analyses (Zar, 1999) were adopted if data were not normally distributed. The Kruskal–Wallis test (Zar, 1999) was used to examine the difference in the mean of the transmission loss coefficient of different test signals running in the sound propagation model. The Mann–Whitney U-test (Zar, 1999) was used to analyze differences between transmission loss coefficients, as well as acoustic parameters between sites. Differences in apparent source levels and energy flux density across different whistle tonal types was analyzed by the Kruskal–Wallis test (Zar, 1999), and Duncan’s multiple comparison test (Zar, 1999) was used for post hoc comparisons of acoustic parameter among tonal types. Statistical analyses were performed using SPSS 16.0 for Windows (SPSS Inc., Chicago, USA). Differences were considered significant at p < 0.05.

Ethical statement

Permission to conduct the study was granted by the Ministry of Science and Technology of the People’s Republic of China. The research permit was issued to the Institute of Hydrobiology of the Chinese Academy of Sciences (Permit number: 2011BAG07B05).

Results

Six hundred and thirty four whistles were recorded during 14 observation days, from which 33 whistles were successfully selected from two days in the Pearl River Estuary and 209 whistles from eight days in the Beibu Gulf (Table 1) for further analysis.

Table 1 Summary of 14 survey days in Pearl River Estuary and Beibu Gulf.

Each successfully localized whistle was grouped according to tonal types.

Site	Date	Sample rate	Recorded whistles			Localized whistles					
				Flat	Down	Rise	U-shape	Convex	Sine	Sum	
PRE	20140605	200,000	78	21	0	2	0	1	5	29	
	20140708	512,821	5	0	0	0	0	0	0	0	
	20140710	512,821	19	1	0	0	0	0	3	4	
	20140711	512,821	6	0	0	0	0	0	0	0	
BG	20140804	512,821	35	4	3	2	3	3	4	19	
	20140805	512,821	49	2	2	1	15	1	2	23	
	20140806	512,821	28	5	5	1	1	0	0	12	
	20140813	200,000	107	6	2	2	1	3	1	15	
	20140814	200,000	55	8	1	4	9	1	0	23	
	20140815	200,000	8	1	0	0	0	0	1	2	
	20140816	200,000	4	0	0	0	0	0	0	0	
	20140820	200,000	66	5	3	2	2	2	9	23	
	20140821	200,000	18	0	0	0	0	0	0	0	
	20140822	200,000	156	13	12	14	33	4	16	92	
	Sum		634	66	28	28	64	15	41	242	

Sound propagation modeling

Sixteen whistles with estimated source distance from one of the 5 hydrophone channels close to 1 m were selected as a proxy for whistle sources and imported to the sound propagation model. The sound propagation speed in water (cw) was calculated as 1,538 m/s and 1,535 m/s, for the Pearl River Estuary and Beibu Gulf, respectively, according to the Medwin equation (Medwin, 1975) based on the average measured site-specific temperature, salinity, and water depth (Table 2). The sound speed in air (cs) was 343 m/s, and in bottom sediment (cb) was 1,535 and 1,742 m/s for Pearl River Estuary and Beibu Gulf, respectively. The impedances zs and zw were 400 and 1.54 × 106 Pam−1 s, zb was 2.19 × 106 and 3.45 × 106 Pam−1 s for Pearl River Estuary and Beibu Gulf, respectively (Urick, 1983), since the sediment types were different, with clay silt in Pearl River Estuary and fine sand in Beibu Gulf (TQ Zeng & HW Su, pers. comm., 2014). The hydrophone depth was set at 1 m to mirror the real hydrophone setup during sound recording. The water depth in Pearl River Estuary and Beibu Gulf was set as a range from 5 m to 9 m and from 2 m to 8 m, respectively, to mirror the measured site-specific depth (Table 2). The maximum distance between animal and hydrophone was set at 50 m (Fig. 4), corresponding to the maximum localized whistle distance of 49.6 m (see next paragraph). The reflective coefficients at the air–water interface were averaged at −0.09 for both of the Pearl River Estuary and Beibu Gulf, whereas the reflective coefficients at the water-bottom interface were averaged at 0.24 and 0.46 for Pearl River Estuary and Beibu Gulf, respectively. No significant difference in the mean transmission loss coefficient k was observed among different testing signals within each site (Pearl River Estuary: Kruskal–Wallis χ2 = 16.82, df = 15, p = 0.33; Beibu Gulf: Kruskal–Wallis χ2 = 5.02, df = 15, p = 0.91). Thus, we pooled all test signals within the sites to calculate average transmission loss. From the pooled data we estimated an average k in Pearl River Estuary of −17.3 ± 1.0 (95% CI [−17.4:−17.2]), which was significantly different from that in Beibu Gulf of −14.6 ± 0.8 [95% CI (−14.7:−14.5)] (Mann–Whitney U-test: z = 1,532, df = 1,119, p < 0.01) (Fig. 9).

Figure 9 Histogram (A) and box plot (B) of the modeling sound transmission loss coefficient in Pearl River Estuary and Beibu Gulf.

In the box plot, the central line mark on each box is the median, the edges of the box are the first quartile (Q1) and the third quartile (Q3), and the notch indicates the 95% CI of the median. Outliers (Open circles) were the data that is outside the fences of Q1 − 1.5 × inter-quartile-range (IQR) and Q3 + 1.5 × IQR, where IQR = Q3–Q1; Whiskers show the most extreme data points that are not outliers.

Table 2 Environmental parameters and the estimated sound propagation speed at the recording site of the Pearl River Estuary and Beibu Gulf.

		Temperature(°C)	Salinity (‰)	pH	Depth(m)	Velocity(m/s)	Sediment	
PRE(N = 61)	Mean ± SD	28.6 ± 0.7	30.6 ± 3.7	7.5 ± 0.4	7.0 ± 0.9	1,538	Clayey silt	
	Range	27.3–30.2	30.1–31.1	6.6–7.9	4.6–9.3			
BG(N = 45)	mean ± SD	30.6 ± 0.4	23.5 ± 6.0	8.1 ± 0.1	4.6 ± 2.0	1,535	Fine sand	
	Range	30.1–31.1	20.0–30.0	7.7–8.3	1.9–7.9			

Acoustic localization

Of all the analyzed whistles, the average estimated distance between the center hydrophone and phonating animals was 6.8 ± 4.2 (SD) m (range: 2.1–23.8 m) in Pearl River Estuary and 8.4 ± 7.0 (SD) m (range: 2.2–49.6 m) in Beibu Gulf (note that the distribution of the distance between hydrophone and phonating animal may vary if a different hydrophone was chosen). The average localization error (εrms) was 0.3 ± 0.2 m and 0.5 ± 0.4 m in Pearl River Estuary and Beibu Gulf, respectively.

Apparent source levels and source energy flux density

Significant differences in whistle duration was observed between Pearl River Estuary (mean ± SD: 0.50 ± 0.19 s) and Beibu Gulf (mean ± SD: 0.44 ± 0.20 s) (Mann–Whitney U test, z = − 2.0, df = 241, p = 0.04) (Table 3). On the other hand, no significant differences were found in all the measured apparent source levels and source energy flux density between Pearl River Estuary and Beibu Gulf (Mann–Whitney U test, p < 0.05) (Table 3), thus they were pooled according to tonal classes for further analysis.

Table 3 Descriptive and comparative statistic of the whistle parameters and active communication spaces from Pearl River Estuary and Beibu Gulf.

	PRE(n = 33)		BG(n = 209)				
	mean ± SD	Range	mean ± SD	Range	z	p	
t	0.50 ± 0.19	0.23–1.01	0.44 ± 0.20	0.21–1.61	−2.0	0.04	
ASLrms	138.5 ± 6.8	125–158	137.2 ± 7.0	114–160	−0.8	0.42	
ASLrms200	140.3 ± 7.3	126–160	139.3 ± 6.9	116–161	−0.4	0.66	
SEFD	135.2 ± 7.4	121–155	134.0 ± 6.8	110–158	−0.7	0.46	
SPLnoi	122.3 ± 5.0	111–134	122.2 ± 6.3	105–132	−1.0	0.32	
fp	4.9 ± 1.0	3.2–8.0	6.1 ± 2.6	1.6–16.0	−3.0	<0.001	
Mean(ACS)*	14.7 ± 2.6	3.6–34.6	34.2 ± 9.5	6.1–76.5	−5.5	<0.001	
Max(ACS)*	17.1 ± 3.5	3.9–39.8	43.5 ± 12.2	7.2–119.2	−5.8	<0.001	
Mean(ACSat)*	24.3 ± 4.8	6.9–65.3	60.7 ± 18.1	6.8–142.6	−4.2	<0.001	
max(ACSat)*	37.5 ± 4.6	7.4–76.5	74.3 ± 25.3	8.1–198.1	−4.4	<0.001	
Notes.

* Denote data with a grossly skewed distribution and descriptive parameters of median, quartile deviation and P5-P95 were presented. Bolded numbers indicating significantly different at p < 0.05.

t whistle duration

ASLs apparent source levels

fp peak frequency derived from a running average of the whistle mean one-third octave band levels with step window size of 200 ms and 95% overlap

ACS the active communication space

ACSat the auditory-threshold limited active communication space

The 242 successfully located whistles consisted of 66 flat, 28 down, 28 rise, 64 U-shape, 15 convex, and 41 sine whistles (Table 1). Significant differences were observed in the whistle duration, apparent source levels, and source energy flux density among the six tonal types (whistle duration: Kruskal–Wallis χ2 = 29.27, df = 5, p < 0.01; ASLrms: Kruskal–Wallis χ2 = 17.02, df = 5, p < 0.01; ASLrms200: Kruskal–Wallis χ2 = 12.08, df = 5, p = 0.03 and source energy flux density: Kruskal–Wallis χ2 = 11.16, df = 5, p = 0.04). In detail, the duration of sine whistles (mean ± SD: 0.60 ± 0.3 s) was significantly longer than flat (mean ± SD: 0.43 ± 0.2 s), rise (mean ± SD: 0.43 ± 0.2 s) (Duncan’s multiple-comparison test; p < 0.05) and U-shape (mean ± SD: 0.39 ± 0.2 s) whistle types (Duncan’s multiple-comparison test; p < 0.01). Sine whistles had significantly lower ASLrms200 andsource energy flux density (Duncan’s multiple-comparison test; p < 0.05) and ASLrms (Duncan’s multiple-comparison test; p < 0.01) than down whistles (Fig. 10).

Figure 10 Box plot of the apparent source levels (ASLs) and source energy flux density (SEFD) of the six tonal types.

The center of each box is the median value, the upper and lower box borders are the first quartile (Q1) and the third quartile (Q3). The whiskers extend to the most extreme data within the fences of Q1 − 1.5 × inter-quartile-range (IQR) and Q3 + 1.5 × IQR, where IQR = Q3–Q1. Outliers (open circles) were the data outside the fences. The boxes with different lower case and upper case were significantly different at p < 0.05 and p < 0.01, respectively, within each apparent source level and source energy flux density categories.

One-third octave band levels

The average peak frequency across the one-third octave band levels for all the 200 ms whistles slices across all whistles was 4.9 ± 1.0 kHz and 6.1 ± 2.6 kHz for Pearl River Estuary and Beibu Gulf, respectively, which were significantly different (Mann–Whitney U test, z = − 3.0, df = 241, p < 0.01) (Table 3). No significant difference was observed in the ambient noise sound pressure levels between Pearl River Estuary (SPLnoi; mean ± SD: 122.3 ± 5.0 dB) and Beibu Gulf (mean ± SD: 122.2 ± 6.3 dB) (Mann–Whitney U test, z = − 1.0, df = 241, p = 0.32) (Table 3). However, their one-third octave band noise level property was significantly varied: in Pearl River Estuary, the one-third octave band level at frequency band of 4.47–5.63 kHz (centered at 6.3 kHz and corresponding to the peak frequency of whistles in Beibu Gulf) was significantly higher than the band of 5.63–7.08 kHz (centered at 5 kHz and corresponding to the peak frequency of whistles in Pearl River Estuary) (mean ± SD: 95.8 ± 3.2 dB and 97.1 ± 4.2 dB, respectively; Wilcoxon signed ranks test, Z = − 4.99, p < 0.001, N = 33), whereas, an opposite trend was observed in Beibu Gulf (mean ± SD: 99.3 ± 4.3 dB and 96.9 ± 4.2 dB, respectively, for the two frequency bands; Wilcoxon signed ranks test, Z = − 8.59, p < 0.001, N = 209). In addition, the one third octave level at frequency band of 4.47–5.63 kHz in Pearl River Estuary was significantly higher than that in Beibu Gulf (Mann–Whitney U-test: z = − 4.02, df = 241, p < 0.001), although no significant difference in the frequency band of 5.63–7.08 kHz was observed between them (Mann–Whitney U-test: z = − 1.68, df = 241, p = 0.09).

Active communication space

We calculated the mean and maximum active communication spaces across the 200 ms segments for each analyzed whistle. The median of these results were 14.7 ± 2.6 m and 17.1 ± 3.5 m, respectively, in Pearl River Estuary, and 34.2 ± 4.8 m and 43.5 ± 4.6 m, respectively, in Beibu Gulf. Both measures were significantly smaller in Pearl River Estuary than in Beibu Gulf (Mann–Whitney U test: z = − 5.5, df = 241, p < 0.01 and z = − 5.8, df = 241, p < 0.01, respectively). The largest mean active communication spaces for any whistle in Pearl River Estuary and Beibu Gulf were estimated to be at 40.7 m and 209.7 m, respectively, while the largest maximum active communication spaces were estimated to be 51.1 m and 266.8 m, respectively. Significant differences were observed in the mean and maximum active communication space among six tonal types, which follow from the observed differences in apparent source level (mean active communication space: Kruskal–Wallis χ2 = 25.56, df = 5, p < 0.01; maximum active communication space: Kruskal–Wallis χ2 = 23.80, df = 5, p < 0.01). In detail, both of the mean and maximum active communication space of flat was significantly shorter than that in down and U-shape (Duncan’s multiple-comparison test; p < 0.01). The mean active communication space of flat was significantly short than that in rise (Duncan’s multiple-comparison test; p < 0.01). Of all the localized whistles, the averaged one-third octave band levels of all matched 200 ms noise samples for each whistle at the frequency determined by the peak frequency of the averaged one-third octave band levels of all the whistle 200 ms slices ((mean(Noi_TOBL))(fp)) were significantly higher than Sousa auditory threshold at the same frequency (AT(fp)) (Mann–Whitney U test, z = − 9.25, df = 241, p < 0.01), indicating that active communication space was mainly noise-limited. However, 62 out of 242 whistles (26.6 %) with the (mean(Noi_TOBL))(fp) lower than its corresponding AT(fp), indicating that their active communication space were auditory-threshold limited, as opposed to noise-limited, we also estimated the theoretical auditory-threshold-limited active communication space for all whistles. The theoretical mean auditory-threshold limited active communication space in Pearl River Estuary (median ± QD: 24.3 ± 4.8 m) was significantly shorter than that in Beibu Gulf (median ± QD: 60.7 ± 12.2 m) (Mann–Whitney U test, z = − 4.2, df = 241, p < 0.01), and the maximum was also significantly shorter in Pearl River Estuary (median ± QD: 35.7 ± 4.6 m) that in Beibu Gulf (median ± QD: 74.3 ± 25.3 m) (Mann–Whitney U test, z = − 4.4, df = 241, p < 0.01) (Table 3). The biggest mean auditory-threshold limited active communication space in Pearl River Estuary and Beibu Gulf were estimated to be 106.5 m and 457.2 m, respectively, whereas the biggest maximum ACSat were estimated to be 109.3 m and 463.8 m, respectively. Significant difference was observed in the mean and maximum auditory-threshold limited active communication space among six tonal types (Kruskal–Wallis χ2 = 17.04, df = 5, p < 0.01 and Kruskal–Wallis χ2 = 15.28, df = 5, p < 0.01, respectively). In detail, both of the mean and maximum auditory-threshold limited active communication space of flat were significantly shorter than that in U-shape (Duncan’s multiple-comparison test; p < 0.05).

Discussion

Although the majority of the analyzed data were from one day of survey for both areas, with 44% (92 out of 209) analyzed whistles in Beibu Gulf from 22 August, 2014 and 88% (29 out of 33) analyzed whistles in Pearl River Estuary from 5 June, 2014, the data within these days was obtained from many different dolphin groups and participated in a variety of activities. Thus, the analyzed data was representative for the region where they were obtained.

Sound propagation modeling

Depending on local conditions, the spreading loss constant k normally ranges from spherical spreading loss (k = 20) to cylindrical spreading loss (k = 10) (Urick, 1983). The transmission loss coefficient in Pearl River Estuary (mean ± SD: −17.3 ± 1.0; range: from −20.5 to −15.5) followed an almost spherical spreading model and was comparable to that calculated for shallow waters of Koombana Bay in Western Australis of −18, which was derived from playback experiments with variable receiver and sender locations at an approximate homogenous water depth of 5–7 m, using pure tone sounds with a frequency span of 1–7 kHz (Jensen et al., 2012) and the empirical and theoretical derived transmission loss coefficient at the Antarctic Peninsula of −17.8 (Širović, Hildebrand & Wiggins, 2007). The transmission loss coefficient in Beibu Gulf (mean ± SD: −14.6 ± 0.8; range: from −16.5 to −12.1) was equidistant between spherical and cylindrical spreading. The range of the transmission loss coefficient estimated in Pearl River Estuary and Beibu Gulf (Fig. 9) was also comparable to that estimated with a water depth less than 3 m and in mud or sand sediment (k range from −26.6 to −14.5) and in channels with water depth between 3 m and 5.3 m (k range from −24.5 to −12.8) in Sarasota Bay, Florida (Quintana-Rizzo, Mann & Wells, 2006). Therefore, our transmission loss estimates were well within the range of currently published findings in similar habitats.

Acoustic localization

The potential factors that influence the source location include ambient noise in the transmission paths, variability in sound speed in the media (Clark & Ellison, 2000; Tiemann, Porter & Frazer, 2004), multipath arrivals of a signal at the hydrophone (Clark & Ellison, 2000; Hayes et al., 2000), and array configurations (Janik, 2000a). However, we estimated low localization error in both of the Pearl River Estuary and Beibu Gulf, with error percentages (εrms divided by mean localized distance between hydrophone and animal (r)) of 0.3/6.8 = 4.4% and 0.5/8.4 = 5.9%, respectively. Therefore, error due to source localization was not expected to be large in our results.

The low localization error in this study may be ascribed to the following counter error analysis strategies, firstly, the use of waveform, rather than spectrogram cross-correlation technique. In order to determine the time delay between two hydrophones, a cross-correlation function can be applied to either signal waveforms or spectrograms (Janik, 2000a). Compared with waveform cross-correlation, spectrogram cross-correlation technique may introduce a slightly larger error (Clark & Ellison, 2000). The error was due to the time grid spacing resolution during spectrogram calculation by averaging over small time slices of the original signals (Janik, 2000a). Secondly, the inclusion of the scanning depth function (an extra dimension to the array plane) in the TOADY localization program in conjunction with the two dimensional array recording systems makes possible the three dimensional localization of the animal and helps to increase the accuracy of the localization (Quick, Rendell & Janik, 2008), since using a two dimensional array in a three dimensional environment may generate some errors (Janik, 2000a). On the other hand, the optimal source location in passive acoustic localization can be achieved when phonation animals inside the array and the source-to-hydrophone distance of the same order of magnitude as inter-hydrophone spacing (Madsen & Wahlberg, 2007). Additionally, source-to-array distance to a range of three to four times the hydrophone array dimension can still be localized with high confidence (Watkins & Schevill , 1972). Of all the localized whistles in this study, those with a source-to-hydrophone distance smaller than 3 m, 12 m, and 15 m, which mimics the distance of one time, three times, and four times of the maximum inter-hydrophone spacing, account for 27.3%, 83.9% and 93.9%, respectively. In addition, the 95% confidence interval of the mean distance between the animal and hydrophone (Pearl River Estuary: 5.4–8.3 m, Beibu Gulf: 7.4–9.3 m) was well within three times of the maximum inter-hydrophone spacing. This also account for the high acoustic localization in this study.

Apparent source level and noise levels

The apparent source levels of Sousa whistle obtained in this study, with the ASLrms over its 98% energy window of 137.4 ± 6.9 dB (range: 114.1–160.4 dB) and ASLrms over 200 ms running windows of 139.5 ± 6.9 dB (range: 115.6–161.4 dB), were similar to the estimated mean and range of the ASLrms of Atlantic spotted dolphin (Stenella frontalis) and bottlenose dolphin at Gulf of Mexico (Frankel et al., 2014), but lower than those found for other dolphin species, including Hawaiian spinner dolphins (Stenella longirostris) (Watkins & Schevill, 1974), common dolphin (Deiphinus delpliis) (Fish & Turl, 1976), white-beaked dolphins (Rasmussen et al., 2006), baiji (Lipotes vexillifer) (Wang et al., 2006), short-finned pilot whale (Globicephala macrorhynchus) (Fish & Turl, 1976) and killer whale (Orcinus orca) (Miller, 2006), and bottlenose dolphin (Fish & Turl, 1976; Janik, 2000a; Jensen et al., 2012; Tyack, 1985) in other regions (Table 4). The observed no significant difference in broad band noise level between the two regions in our study may be due to the fact that, in order to model the active communication range of whistles in real time noise conditions, the analyzed noises were derived from sound files with whistle recorded in good SNR and successfully localized. However, the majority of good whistles were obtained either far away from the construction or navigation region in Pearl River Estuary or without dolphin tourism boat nearby in Beibu Gulf and represent an environment with less anthropogenic noise pollution. Besides, the fact that the successfully localized whistles in this study were from a less anthropogenic impacted environment may, in part, account for the low source level of the humpback dolphin whistles, since dolphins may use higher amplitude sound in a noisy environment. Additionally, the frequency of dolphin whistles tend to modify according to environmental ambient noise, and the bottlenose dolphin was observed to produced whistles with lower (Morisaka et al., 2005) or higher (May-Collado & Wartzok, 2008) frequencies at higher ambient noise conditions. In the present study, both the averaged peak frequency of whistles and one third octave band level of noise were significantly different between Pearl River Estuary and Beibu Gulf. However, the reasons why humpback dolphins emitted whistles with peak frequency coincidence with the noise one third octave frequency band with higher noise level in both of these two regions deserve further investigation.

Table 4 Apparent source level of dolphin and whale whistles.

Species	Latin name	Source	Location	Mean ± SD (dB)	Range (dB)	Sample size	
Hawaiian Spinner dolphins	Stenella longirostris	Watkins & Schevill (1974)	Kealakekua Bay, Hawaii		109–125	N = 14	
		Lammers & Au (2003)	Coastal of Oahu, Hawaii ar	153.9 ± 4.5a		N = 22	
				150.2 ± 2.8b		N = 22	
Bottlenose dolphin	Tursiops truncatus	Tyack (1985)	Captive in aquariums, US		125 ≥ 140c		
		Fish & Turl (1976)	Offshore Southern California waters		150–173d		
		Janik (2000a)	Moray firth, Scotland	158 ± 6.4	134–169	N = 103	
		Jensen et al. (2012)	Koombana Bay, Western Australia	146.7 ± 6.2e	136.8–158.0f	N = 180	
				147.6 ± 6.4g	137.9–159.0f	N = 180	
		Frankel et al. (2014)	Gulf of Mexico	138.2 ± 8.0	114–163	N = 645	
Common dolphin	Deiphinus delpliis	Fish & Turl (1976)	Offshore Southern California waters		125–145h	N = 385	
Atlantic spotted dolphin	Stenella frontalis	Frankel et al. (2014)	Gulf of Mexico	138.4 ± 8.0	115.4–163.1		
White-beaked dolphins	Lagenorhynchus albirostris	Rasmussen et al. (2006)	Faxafloi Bay, Iceland	148 ± 12i	124–166	N = 12	
				144 ± 8j	118–167	N = 43	
Baiji	Lipotes vexillifer	Wang et al. (2006)	Shishou reserve, China	143.2 ± 5.8	135.9–150.8	N = 43	
Humpback dolphin	Sousa chinensis	Present study	PRE and BG, China	137.4 ± 6.9k	114–160	N = 242	
				139.5 ± 6.9l	116–161	N = 242	
Short-finned pilot whale	Globicephala macrorhynchus	Fish & Turl (1976)	Offshore Southern California waters		157–183d		
Killer whale	Orcinus orca	Miller (2006)	Johnstone strait, Canada	140.2 ± 4.1	133–147	N = 24	
Notes.

All source levels were in rms, except those from the study of Fish & Turl (1976), of which power spectrum range were given.

a Animal moving with or towards the array

b Animal moving ahead of or away from the array

c Limited by the recording system

d Measured from power spectrum generated by peak hold method

e Measured from 95% energy window

f 90% CI

g Measured from 200 ms running windows

h Measured from power spectrum generated by sum-average method

i Measured from 100 ms running window

j Measured by cross-correlation functions

k Measured from the 98% energy window

l Measured from 200 ms running window

Active communication space

Active communication space has been estimated for bottlenose dolphins (Janik, 2000a; Jensen et al., 2012; Quintana-Rizzo, Mann & Wells, 2006), killer whale (Miller, 2006), white–beaked dolphins (Rasmussen et al., 2006) and baiji (Wang et al., 2006). In these studies, the whistle source levels used to model the active communication space were either the highest and lowest source levels (Rasmussen et al., 2006), the mean source level (Wang et al., 2006), the highest and mean source levels (Janik, 2000a), the three ‘representative’ source levels (Quintana-Rizzo, Mann & Wells, 2006), the source level of each whistle over its 95% energy window (Jensen et al., 2012) or on the one-third octave band level scale (Miller, 2006). The noise condition was either extrapolated from literature at other habitats at sea-state 0 and sea-state 4 or 6 (Janik, 2000a; Miller, 2006), or using the noise level at the habitat in question under an optimal conditions of sea-state 0–1 and without an adjacent boat (Jensen et al., 2012; Rasmussen et al., 2006), or under normal recording conditions (Quintana-Rizzo, Mann & Wells, 2006; Wang et al., 2006). The propagation model was either adopted from the spherical transmission attenuation model (Rasmussen et al., 2006; Wang et al., 2006), the Marsh and Schulkin shallow-water transmission model (Marsh & Schulkin, 1962) which assumes the transducer and receiver were in the middle of the water column (Janik, 2000a; Miller, 2006) or derived by the sound transmission experiment using playback signal of computer-generated whistle mimic tone at varied distance with the transducer and receiver at a fixed depth of 1 m (Quintana-Rizzo, Mann & Wells, 2006), or with varied transducer and receiver combination (Jensen et al., 2012). However, when using the signal ASLrms for modeling active communication space, the critical band theory of the mammalian auditory system was not considered, and may therefore have integrated the energy from harmonics, which was the integer times of the fundamental frequency and tends to be directional (Lammers & Au, 2003; Miller, 2002; Rasmussen et al., 2006), and more heavily affected by absorption. Additionally, the optimal or averaged noise condition, rather than the real-time ambient noise level used for modeling active communication space, may not shed much light for the biologically relevant active communication space of the signal. This was further corroborated by the findings that the modeled active communication space of baiji whistles with a ASLrms of 143.2 dB will reduced from 6.6 km at normal noise level to a range of 22–220 m in a noisy boat traffic conditions. Furthermore, the Marsh and Schulkin model or playback experiment with the transducer and receiver at a fixed depth may not represent the transmission loss pattern of the site, because the sound transmission loss may vary within the water column (Quintana-Rizzo, Mann & Wells, 2006). Transmission loss near the surface or the bottom of the water column was observed to be much higher than that at the center of the water column (Janik, 2000a). One-third of an octave approximates the effective filter bandwidth of mammalian hearing systems (Fletcher, 1940; Richardson et al., 1995). The signal one-third octave band level information provided us with a starting point and an appropriate way for investigating how a mammal’s auditory system perceives sound and the extent of the masking effect of the ambient noise (Blackwell, Lawson & Williams, 2004; Madsen et al., 2006). In this study, models integrating the signal one-third octave band levels for each individual whistle, its corresponding real-time noise conditions, and site-specific transmission attenuation were adopted to estimate the active communication space in a more meaningful way. It should be noted that sound detection and discrimination thresholds may differ; for example, in all the birds of budgerigars (Melopsittacus undulates), zebra finches (Taeniopygia guttata) and canaries (Serinus canaria), the thresholds for discrimination between calls of the same species was observed higher than the thresholds for detection of those calls (Lohr, Wright & Dooling, 2003). The weakest portions of an emitted signal will always be lost during transmission before it is detected by a receiver, but whether or not the transmitting animal can be discriminated by a receiving animal will depend on how much of the information in the signal is needed to solve this task. Individual discrimination information of bottlenose dolphins was encoded in the frequency modulation contour of their signature whistles (Janik, Sayigh & Wells, 2006). Thus, the active communication space of signature whistles was determined by the weakest portion of the signal modulation contour. However, whether signature whistles also exist in humpback dolphins is still unknown and deserve further research. Since whistles were narrowband signals, we believe that the active communication space calculated by using the mean(Sig_ TOBL), as obtained by a running average of all the one-third octave band levels for each whistle over its 98% energy windows, will be similar to that based on the bandwidth of the entire whistle.

The small active communication space calculated in this study is a result of ambient noise limiting detection ranges. The averaged one-third octave band noise levels were 98.7 ± 4.5 and 96.8 ± 4.2 dB re 1 µPa at the band which accommodates the peak frequency of whistles at 4.9 kHz and 6.1 kHz for Pearl River Estuary and Beibu Gulf, respectively. These results were almost 20 dB higher than the one-third octave band noise levels at the same frequency band (estimated to be less than 75 dB) for the estimation of the whistle communication range of the white-beaked dolphins in Faxafloi Bay, Iceland (Rasmussen et al., 2006), almost 30 dB higher than the ambient noise level (estimated to be 66.8 dB) for the estimation of the whistle communication range of baiji in the Shishou reserve (Wang et al., 2006), about 50 and 30 dB higher than the ambient noise level at sea state 0 and 4 respectively used for the estimation of the whistle communication range of bottlenose dolphins at Moray Firth (Janik, 2000a), and almost equal to the background noise spectrum level plus critical ratios for the estimation of the whistle communication range of bottlenose dolphins in Sarasota Bay (Quintana-Rizzo, Mann & Wells, 2006). The high ambient noise level observed in both of the Pearl River Estuary and Beibu Gulf at less anthropogenic impacted areas may due to the waves and biological noise, such as snap shrimps, and deserves further research.

The flourishing year round dolphin-watching industry makes the Beibu Gulf dominated by dolphin-watching boats of about 7 m length and equipped with 40 horse power engine during the day time, whereas the fast developing local economy makes Pearl-River Estuary dominated by different kinds of hydrofoil ferries between Hong Kong, Zhuhai, and Macao throughout the day, with the ferry length of 27.4–47.5 m and speed of 28–45 knots (Z-T Wang, 2014, unpublished data). Since different vessels tend to had very different acoustic signatures (Hermannsen et al., 2014). The varied vessel traffic condition between these two regions might also be a clue to the sources contributing to the ambient noise. Longtime noise recordings will be needed for a better view of the soundscape difference between these two regions.

The apparent source levels between different tonal classes was only significant varied between sine and down. The significant difference in active communication space between flat and down, and between U-shape and rise may be due to their frequency band variations (Wang et al., 2013), since the active communication space was determined by the one-third octave band level at the whistle peak frequency and the corresponding noise one-third octave levels at the same frequency band. In addition, the significant differences in apparent source levels and its active communication space among different whistle tonal types may be associated with their different functions in different behavioral context. This was further corroborated by the findings that different whistle tonal classes were in relation with different behavior states in Indo-Pacific bottlenose dolphins (Hawkins & Gartside, 2010) and some whistle types were used to convey specific information on their specific behavioral context (Hawkins & Gartside, 2010).

Acoustic cues play an important role in mediating social structure and were critical for aquatic animals, especially cetaceans that rely heavily on acoustic for communication. The active communication space calculated in this study can be used for defining dolphin groups in a more biologically meaningful way during field surveys. As determined by the mean active communication range, humpback dolphins within the distance of 14.7 m in Pearl River Estuary and 34.2 m in Beibu Gulf might be able to keep acoustic contact and can be defined as the same dolphin group. In a more quieter environment, where dolphin auditory threshold determine the active communication range, animals which apart from each other at a distance of less than 24.3 m in Pearl River Estuary and 60.7 m in Beibu Gulf can be grouped into the same dolphin group.

Auditory masking of communication signals may interfere the acoustically mediated social interactions. During our field survey in Beibu Gulf, when fast moving dolphin-watching boat was presented even at a distance of more than 1,000 m away, the whistles recorded will be severely masked with the generated vessel noise dominating the ambient noise. The negative impacts of vessel noise on cetacean have been widely documented. The vessel noise recorded at the heavily ship-trafficked marine habitats in Denmark from a range of different ship types was observed substantially elevated ambient noise levels across a wide frequency band of 0.025–160 kHz and can cause hearing range reduction of over 20 dB on harbor porpoises (Phocoena phocoena) even at a distances of over 1 km away (Hermannsen et al., 2014). Small vessels noise can significantly mask acoustically mediated communication in cetaceans and those travelling at 5 knots in shallow waters of the Koombana Bay, Western Australia can reduce the communication range of 26% on bottlenose dolphins within a distance of 50 m (Jensen et al., 2009a). The impact of vessel traffic on local humpback dolphins, especially the hydrofoil ferries at heavy shipping lanes of the PRE, deserves further research.

Limitations

Cetacean audiograms can vary greatly among individuals of different age groups (Houser, Gomez-Rubio & Finneran, 2008; Popov et al., 2007), and the threshold may shift with the presence of masking noise (Johnson, 1968a). Therefore, the audiogram adopted in this study may not be considered representative of the auditory sensitivity of Sousa as a whole, and more auditory studies covering different dolphin age groups are needed for deriving an accurate audiogram of Sousa. The source level of dolphin whistles can vary depending on the number, age, and sex of the individuals in a group, as well as their behavioral context (Fish & Turl, 1976; Frankel et al., 2014). The maximum apparent sound pressure level of 160.4 dB (rms) observed here does not necessarily represent the capability of the species. Due to the trade-off between high SNR and increased active space for high amplitude sound and decreased detection probability by predator with low amplitude sound (Morisaka & Connor, 2007), dolphins might not produce whistles at their maximum levels, even if rewarded in a conditioning procedure (Janik, 2000a).

The directional pattern of outgoing signals and any directional hearing capability will impact the active communication range of animal vocalization (Au, 1993). Received sound pressure levels will be maximum when the phonating dolphin is pointed directly at the receiver, and vice verse. In this study, both the directivity index (DI) and the beam pattern (BP) of the outgoing signal transmission and sound receiving at the frequency of the Sousa whistle were assumed to be 0 dB. The assumption of the 0 dB directivity index of the sound receiving system was further corroborated by the findings that bottlenose dolphins’ sound receiving directivity index increased with frequency and followed the equation (Au, 1993): (18) DIreceiving=16.94×log10f−14.69

where DI was directivity index, and f was frequency in kHz. If this was also applicable and can be extrapolated for Sousa, then the directivity index for signals with frequencies lower than 7.37 kHz will be close to 0.

Previously, 4 cm and 6 cm radius circular piston projectors were applied to model the outgoing signal transmission beam pattern for spinner dolphin (Lammers & Au, 2003) and white-beaked dolphin whistles (Rasmussen et al., 2006), respectively.

The same method (Au, 1993; Wang et al., 2015) was adopted in this study, and the modeled beam pattern has a directivity index of 3 dB for a 4 cm circular piston model and 6 db for a 6 cm circular piston model (Fig. S2), suggesting directionality would also have a minor effect on the results reported here. These are only theoretical results, however, and need to be further corroborated by empirical measurements.

Conclusions and implication of the findings

Regulation of anthropogenic noise is often limited by insufficient data on marine mammal vocalization characteristics and on the impacts of noise on their biological functions. The present study back-calculated the apparent source levels of free-ranging Indo-Pacific humpback dolphin whistles in the shallow-water environments of Pearl River Estuary and Beibu Gulf, China, by using a 2-dimension cross-shaped hydrophone array system to localize phonating animals and site-specific sound propagation modeling. The dolphins produced whistles with a mean apparent source levels of 138.5 ± 6.8 dB and 137.2 ± 7.0 dB re 1 µPa (rms) for Pearl River Estuary and Beibu Gulf, respectively. The corresponding mean and maximum active communication space of Sousa whistle were estimated with a median of 14.7 m and 17.1 m for Pearl River Estuary, and 34.2 m and 43.5 m for Beibu Gulf. Whereas the auditory threshold determined mean and maximum active communication range were estimated with a median of 24.3 m and 35.7 m for Pearl River Estuary, and 60.7 m and 74.3 m for Beibu Gulf by integrating the real-time ambient noise levels, masking tonal threshold, Sousa auditory threshold and site-specific transmission loss model. The present study contributes relevant quantitative baseline data on Sousa vocalization characteristics. Since the broadband anthropogenic noise tend to overlap in frequency range with the acoustic signals of a wider range of marine fauna, the source level information of the dolphin whistles can also be referenced as the safe biological ambient noise that an animal is exposed to and can shed some light for evaluating the appropriate noise exposure level for humpback dolphin and regulation or mitigation of underwater acoustic pollution. Furthermore, although biosonar clicks are easy to detect automatically at close distance, and standard methods have been developed for passive acoustic monitoring of biosonar clicks (Zimmer, 2011), whistles may be detected at a greater range compared with the biosonar clicks, which are higher in directionality and suffering higher transmission loss. The whistle apparent source levels and site-specific transmission loss model derived in this study can aid in expanding the application of passive acoustic monitoring strategies. These include the scope of local population abundance estimation by incorporating the active detection range of the passive acoustic recorders (Wang et al., 2014a), propagation characteristics of the environment, and animal vocalization rates (Frankel et al., 2014). Finally, the active communication space calculated in this study can be used to determine how far apart members of dolphin group might be able to keep acoustic contact and be used for defining dolphin groups in a more biologically meaningful way during field surveys. In addition, it can guide the appropriate approach distance for local dolphin-watching boats and research boat during focal group following.

Supplemental Information

Figure S1 Power spectral density of ambient noise in Beibu Gulf at sea state of zero and hydrophone self-noise

The nominal self-noise level was obtained when the hydrophone was connected to the voltage pre-amplifier VP2000 and was provided courtesy of the Reson company. The gray lines were the 95% confidence interval of the ambient noise.

Click here for additional data file.

File S1 Sound propagation model

Click here for additional data file.

Figure S2 Beam pattern of circular piston transducer

Modeling was based by using a radius of 4 cm (directivity index = 3 dB) and 6 cm (directivity index = 6 dB) piston transducer at typical Sousa whistle frequency of 6.35 kHz.

Click here for additional data file.

The hydrophone array system was provided courtesy of the Public Technology Service Center, IHB, CAS. We gratefully acknowledge the captains of Hongwei Su, Shaode Huang, Hongyu Su, Hongfang Su, Yingqiang Li for their patience and cooperation, and Xiangxia Wu and Rui Zheng for logistic support. Individual thanks are due to Wenjun Xu of the Ningbo No. 2 High School of Zhejiang province for her field and statistical assistance, Paul E. Nachtigall and Marc O. Lammers of the Hawaii Institute of Marine Biology of the University of Hawaii for their helpful discussion about this study. Special thanks are also extended to Susan Schmidt and the academic editor and many reviewers for their helpful critique of an earlier version of this manuscript.

Additional Information and Declarations

Competing Interests

Author Contributions

Field Study Permissions

Data Availability

The authors declare there are no competing interests.

Zhi-Tao Wang conceived and designed the experiments, performed the experiments, analyzed the data, contributed reagents/materials/analysis tools, wrote the paper, prepared figures and/or tables, reviewed drafts of the paper.

Whitlow W.L. Au and Luke Rendell contributed reagents/materials/analysis tools, wrote the paper, reviewed drafts of the paper.

Ke-Xiong Wang and Ding Wang conceived and designed the experiments, performed the experiments, wrote the paper, reviewed drafts of the paper.

Hai-Ping Wu, Yu-Ping Wu, Jian-Chang Liu, Guo-Qin Duan and Han-Jiang Cao wrote the paper, reviewed drafts of the paper.

The following information was supplied relating to field study approvals (i.e., approving body and any reference numbers):

Permission to conduct the study was granted by the Ministry of Science and Technology of the People’s Republic of China. The research permit was issued to the Institute of Hydrobiology of the Chinese Academy of Sciences (Permit number: 2011BAG07B05).

The following information was supplied regarding data availability:

Figshare: 10.6084/m9.figshare.2068023.v1.

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
