# Peer review of "Apparent source levels and active communication space of whistles of free-ranging Indo-Pacific humpback dolphins (Sousa chinensis) in the Pearl River Estuary and Beibu Gulf, China"

_PeerJ, doi:10.7717/peerj.1695_

## Round 0.1 · original submission · Major Revisions

As you will see below, both reviewers returned signed comments, and both were very positive about the contribution of this ms. to the literature. Regarding the revision, I am not concerned about the overall ms. length, but please pay careful attention to correcting the English, as indicated by the second reviewer.

·

Basic reporting

This is a thoroughly written and coherent ms that I recommend for publication pending revision.
The introduction is well structured and provides sufficient background for the study and discussion.
The ms is very detailed, which makes it easy to understand the methods step by step, but also makes the paper somewhat long (~9500 words from introduction to conclusion). In my opinion the length could easily be reduced by a couple of pages, but since PeerJ is a journal covering numerous biological topics and is not specifically aimed at acousticians then the degree of detail may be justified.
The sentences are generally easy to understand, but there are a few grammatical errors here and there especially in some sections of the discussion which I suggest the authors take a second look at.

Experimental design

The experimental procedures are sound and I only have a few comments or objections that can be considered major (see Comments for the Author). The methods for obtaining the data are within common practice of the field and sufficient information is given that others could reproduce the study. It is my opinion that this study qualifies as being rigorous and of high technical standard.

Validity of the findings

The data are robust and that the discussion is firmly based on the actual findings and does not suffer from overextrapolations.

Additional comments

The authors have done a very thorough job and the methods used are in my opinion of very high standard. I commend the authors for going through the effort of creating a model to estimate the transmission loss coefficient instead of just assuming a number between 10 and 20.
However, I do have some points of critique that must be addressed before I can recommend publication.

Major comments:

1. Since this study only focuses on low-frequency (<10 kHz) whistles, then why was an insensitive TC-4013 hydrophone chosen for the recordings that invariably creates a high self-noise? I question whether the self-noise is below the ambient noise levels? In line 170-172 you report your system noise levels. How were they measured? Was it done by placing the entire recording chain in a silent location or was it merely the noise levels from the analog-to-digital converter itself? Please clarify this in the manuscript. If it was not done for the entire recording chain including the hydrophones, then I ask that you do another system noise measurement and that you plot this result in Figure 8 to show the reader what the relationship is between the reported ambient noise levels and the self-noise to avoid the suspicion that your high ambient noise levels indeed are self-noise levels.

2. Line 203: An SNR criterion of only 5 dB will impose large errors (3-4 dB I suspect) on received level measurements and localization. Since most of the whistles are recorded from a range closer than 15 meters, would it then change the sample size dramatically if the SNR criterion was increased to say 10 dB instead? I suggest that you take a few representative whistles having a high SNR and then add varying degrees of white noise, while computing the RMS levels and do the acoustic localization. It would then be possible to investigate what effects varying degrees of SNR will have on acoustic localization range / source level estimates and then I suggest that you either provide the results as a figure or explain them in the main text.

3. In terms of the active communication space estimations, the data indicate that the dolphins’ possible communication ranges are often limited to a few tens of meters due to the high ambient noise levels even though you made your recordings in relatively “quiet” areas. I do miss a more in-depth discussion of what biological consequences this might have for the animals in terms of group coherence. What I also suggest is to take a few random whistle examples and then model the effects in terms of reduced communication ranges when simply adding varying degrees of white noise (or e.g. the TOL measures of boat noise at varying boat speeds in Figure 3 in Jensen et al 2009b could be used). By doing that it will then be possible to speak about the consequences of reduced communication ranges (the range reduction factor; see Møhl 1981, Jensen et al 2009b, or Hermannsen et al 2014) when the animals are exposed to added noise levels in the presence of whale-watching boats or when close to heavy shipping lanes. Your paper could then include suggestions for maximum advisable approach speeds and approach behaviours for whale-watching boats in the area.


Minor comments and suggestions:

1. Introduction: Given the similarity in objectives between this study and the problems addressed by Jensen et al 2009b I think it would be appropriate to include a reference or two to that study in a suitable context in the introduction. I did see that the Jensen et al 2009b study is cited in line 115, but only with reference to the definition of a specific source parameter. Also, Jensen et al 2012 is cited a lot in the remainder of the paper, but is absent when outlining the background for the study in the introduction. I also suggest to include Clark et al. 2012 “Acoustic masking in marine ecosystems: intuitions, analysis, and implication” in either the introduction, discussion or both.

2. Line 130 and line 137-138: Is it necessary to define what a group of dolphins is, when the only thing that seemingly matters for the recording procedure is whether or not the nearest animal in the area is closer or further away than 50 m? Also, in your introduction where you discuss group definitions, I suggest that you include a sentence or two about why such definitions are important.

3. Line 143: How many poles did the pass band filter have? In order to avoid aliasing problems the low pass filter should not be set on the Nyquist frequency, but a few tens of kHz below depending on the sharpness of the filter and the sampling rate of course.

4. Line 165-168: What was the resulting clipping level of the entire recording chain? It would have been neat if a calibration had been done for the entire recording chain in the field by playing out whistles at known ranges.

5. Line 185: Since you state in line 181 that EFD is on a dB scale then I assume that both TL, EFDr and EFDs in equation 1 are on a dB scale as well. If that is the case then a subtraction and not a division should be used since this gives the ratio on a log-scale.

6. Line 214-215: Are there good reasons to include 3 references by Quick et al. and 2 references by Schultz et al. here? Otherwise I suggest that you just select the most appropriate reference(s) since 6 references here is a little over the top in my opinion.

7. Line 244-247 + Figure 5: Why not include the information given by the fifth/centre hydrophone when doing the acoustic localization so you obtain four independent hyperbolas instead of three? Your localization errors are very small and your range criteria are rather strict so I do not see this as any major issue, but since you have extra time-of-arrival-difference information, then why not use it? At least I suggest that you write why you chose to only use four hydrophones instead of all five.

8. Line 270: Reflections and reverberation may be pronounced in a shallow water environment, so in my opinion it is much cleaner to extract noise information preceding the signal of interest instead of using a recording sequence immediately after the signal.

9. Line 337: Important correction: The original equation reads CR (on dB scale) = a + b*f^(1/2). It makes a substantial difference whether you divide the frequency by 2 or whether you raise it to the power of ½.

10. Line 369-370: Why not use the mean or median source level value of all whistles instead? Only using some whistles localized to ~1 m may be a little dangerous since when acoustic localization goes bad e.g. due to poor SNR, then the estimated ranges will generally be underestimates and may result in very small range estimates even if the true ranges were many times larger. If all sixteen whistles have a high SNR and all hyperbola crossings for the acoustic localizations cross close to each other and thus imply a robust localization, then I have no objections to only using a subset of the whistles.

11. Line 532-534: Here there seems to be some confusion between source level and pitch that to me reads as if they are the same. Source level has to do with the measured amplitude whereas pitch relates to the frequency content (or the perceived sensation of the frequencies) of a sound.

12. Line 537-539: If you mean the third octave band level of the noise centred on the peak frequencies of the whistles then I suggest that you clarify that. Otherwise the sentence may seem a little confusing.

13. Line 539-541: This sentence is also confusing. I suggest rewriting it to clarify what is meant here.

14. Line 574: What is meant by “an almost first approximation”?

15. Line 580-581: The weakest portions of an emitted signal will always be lost during transmission before it is detected by a receiver, but whether or not the transmitting animal can be recognized as a specific individual by a receiving animal will depend on how much of the information in the whistle is needed to solve this task. Maybe the weakest part of a signature whistle is not necessary to be able to do that. I suggest rewriting this sentence and including a small discussion of the difference between detection and discrimination.

16. Line 587: “dB” should be replaced with dB re. 1 µPa. Using dB without a reference is appropriate when comparing different levels, but is meaningless when used about a certain level.

17. Line 593-594: I assume that the order of 30 and 50 should be swapped in order for those numbers to correspond with the correct sea state noise levels.

18. Line 626: Again there seems to be some confusion about source level and pitch. If the frequency/pitch of a sound is increased, then the active space will decrease due to greater absorption at higher frequencies.

19. Line 630-631: The paragraph following “e.g.” is confusing. Please rewrite to clarify the intended point.

20. Line 635: Could it not be imagined that the lowest source level would be recorded when the animal is facing directly away from the hydrophone. Or that dips in the frequency spectrum at certain off-axis angles < 90 degrees may result in lower recorded amplitudes than those recorded exactly at 90 degrees? Also, what is really estimated in this study is not an active space as such, but an active communication range over which an animal whistling at an unknown orientation can be detected when recorded from that specific angle. At other angles the maximum communication range might be different and hence the active communication space cannot be reliably estimated from a recording from a single location even though the transmitting directivity index is likely to be rather low.

21. Line 664-666: What is meant by an animal perceiving the vocalizations of its conspecifics as biological noise? Since information is encoded in the whistles, then these signals are clearly not just biological noise for the animals?

22. Line 915 / Table 3: Reporting source levels to fractions of a dB may be OK when reporting mean or median values, but for the ranges of individually measured values I suggest to round off the numbers both here and throughout the entire manuscript. According to Urick (1983) it is futile to report measures in fractions of a dB as no one can calibrate their equipment to that precision.

23. Line 921 / Table 4: I strongly suggest that you report what kind of source level measures you are referring to. The table legend states what windows were used for the source level calculations, but is it peak, peak-peak, RMS or energy flux density source levels that are measured in those windows? I assume it is one of the latter two? Since the window lengths used are not the same and are based on different criteria, then the RMS-measure may be somewhat problematic and the EFD measure would be preferred. If this table is a mixture of RMS and EFD values then this should be clearly indicated so that apples are not compared with pears.

24. Figure 3: First line of the figure legend. I assume the order of “receiving hydrophone” and “animal” should be swapped.

25. Figure 4: In the Y-axis label please state the reference value of the transmission loss (dB re. 1 m).

26. Figure 6 and Figure 7: According to your methods you used a sampling rate of 200 kHz, and hence the Nyquist frequency must be 100 kHz and not 200.

27. Figure 7B: At approximately 11 kHz there seems to be a constant tone which I suspect is system noise. Did you correct your recordings for this tone when calculating RMS/EFD levels?

28. Figure 10: You cannot report RMS and EFD measures together with the current y-axis. I suggest that you simply use “Source levels” as the y-axis label, and then in the legend in the top right corner include the correct units for each of the three measures.

29. The font sizes of the different figure axes and axes labels vary a bit from figure to figure. I suggest that the font sizes are changed to resemble those used in Figure 7, which makes it a lot easier to read at a distance if for example the figures are to be used in power point presentations etc.

·

Basic reporting

This manuscript meets all the requirements outlined for this section of the review. See General Comments for the Author regarding the need for improvements to the English of the manuscript.

Experimental design

The experimental design is exemplary. Very few articles that consider source levels and active communication space show such thorough experimental design. I hope this will become the standard against which other studies are designed and evaluated.

Validity of the findings

Varying definitions giving different results are used at different points in the manuscript under the same headings of Mean Active Communication Space, Maximum ACS based on existing acoustic conditions and Mean ACS and Maximum ACS based on hearing thresholds. In the Abstract, the Max ACS and Max ACS(at=acoustic threshold) are based on the one whistle type that had the largest source level whereas in the body and in Table 3 these values are given based on all whistle types. This distinction is not at all clear in the manuscript.

Some of the other inconsistencies could be attributed to typographical errors, but not all. It appears that with this many authors, different sections were written at different times and based on different stages of the analysis. See table below and in attachment as it appears in the preview mode that the table structure is lost and it is difficult to align the columns (there are no data for Maximum.ACS and Maximum ACS(at) in the Conclusions).

Abstract Body Conclusions Table 3
Pearl River Estuary
Mean ACS 14.7 14.7 114.7* 16.9
Maximum ACS 51.1 17.1 18.5
Mean ACS(at) 25.3 24.3 24.3 27.5
Maximum ACS(at) 107.3 35.7 31.8

Beibu Gulf
Mean ACS 34.2 34.2 34.2 37.9
Max ACS 266.8 43.5 45.8
Mean ACSat 52.7 60.7 60.7 55.2
Maximum ACSat 463.8 74.3 70.7
*Likely a typo for 14.7

Additional comments

I know that English is not the first language of all the authors but it is of some. Those in the latter category need to do a thorough read and correct the numerous examples of lack of agreement between subject and verb, misuse (or non-use) of the definite article, and general infelicitous expressions. I will highlight some row numbers that particularly need attention, but first a broader point.

The Introduction makes a valid point about the arbitrary nature of the 10 m chain rule or the 100 m radius for identifying group membership. It goes on to note that looking at active communication space is a biologically more meaningful way. The Discussion simply says that the results in this paper can help to identify groups in biologically meaningful ways but does take the data presented and use them to lay out measures of group membership for the dolphins in the two different areas.

Lines to look at for English improvement include: 135, 148 (also here the expression 1MS/s is used. I expect this means 1 million samples/sec, but to be consistent with other terminology and terminology more broadly used, it should be a sampling rate of 1 MHz), 231, 383, 384, 389, 445-451 (can be rewritten to be clearer-no need to put all ideas in one extended sentence), 464, 470, 471, 509-512 (reword for enhanced clarity), 527, 531 (“bout” should read “boats”), 534 (“higher pitch sound” should read “higher level sound”), 539, 551, 559, 561, 562, 565, 570, 571 (strike “formerly” this could be said about everything referenced from published literature and is not needed; “formerly observed much higher” should read “observed to be much higher”), 572 (“bandwidth of mammalians” should read “bandwidth of mammalian hearing systems”), 573 (break multi-line sentence into two by inserting a period after the citation), 581 (“weather” should read “whether”), 582, 598 (“environment” should read “areas”), 606-611 (reword for clarity), 611 (additional grammatical changes in next sentence), 612, 614, 632, and 636.

---

## Round 0.2 · accepted · Accept

The authors have clearly put a great deal of effort and thought into their responses to the comments of the reviewers, both of whom had provided complimentary reviews, and all of this information is part of the publication file.